# Structure and Properties of High-Entropy Boride Ceramics Synthesized by Mechanical Alloying and Spark Plasma Sintering

**DOI:** 10.3390/ma16206744

**Published:** 2023-10-18

**Authors:** Nikolay Razumov, Tagir Makhmutov, Artem Kim, Anatoliy Popovich

**Affiliations:** Institute of Machinery, Materials, and Transport, Peter the Great Saint Petersburg Polytechnic University, Politechnicheskaya Str. 29, 195251 Saint Petersburg, Russia; n.razumov@inbox.ru (N.R.); artyomkim1996@yandex.ru (T.M.); popovicha@mail.ru (A.P.)

**Keywords:** ceramics, high-entropy borides, thermal-oxidative resistance, mechanical alloying, spark plasma sintering

## Abstract

This manuscript shows the study of the structure, mechanical, and chemical properties of high-entropy borides MeB_2_ (Me = Ti, Ta, Nb, Hf, Zr). High-entropy borides were synthesized by mechanical alloying and spark plasma sintering. A chemically homogeneous powder with a low iron content (0.12%) was obtained in a planetary mill by rotating the planetary disk/pots at 200–400 rpm and a processing time of 7.5 h. The structure, mechanical, and chemical properties of the resulting high-entropy borides have been studied. A single-phase hexagonal structure is formed during spark plasma sintering of mechanically alloyed powders at 2000 °C. The microhardness of the samples ranged from 1763 to 1959 HV. Gas-dynamic tests of the synthesized materials showed that an increase in the content of Zr and Hf in the composition increases the thermal-oxidative resistance of the material.

## 1. Introduction

Increasing the temperature range of operation and energy load of machines and mechanisms requires the creation of new generation materials. Such materials should have an optimal combination of mechanical, thermophysical, and chemical properties: high hardness, thermal conductivity, crack resistance, oxidation and corrosion resistance in a wide temperature range, low friction coefficient, and phase stability at elevated temperatures [1,2].

Most studies in this area are devoted to ZrB_2_, HfB_2_ materials, and composites based on them [3,4,5,6,7,8,9]. Hafnium diboride has the highest melting point (3523 K) among borides, high thermal conductivity, hardness and bending strength in a wide temperature range. This applies both to single-phase hafnium diboride and to composite materials based on it. However, the mechanical properties of HfB_2_ significantly depend on structure and porosity [7,8,9,10].

Ceramics based on high-entropy alloys (HEA) is a new class of materials of interest to the world scientific community. In recent years, there has been growing interest in the development of multicomponent and boride ceramics. High-entropy ceramics show better hardness, wear resistance, and oxidation resistance than metal monocarbides and diborides [1,2,11,12,13,14,15,16,17,18,19]. High-entropy borides (HEB) can be used as working materials at ultrahigh temperatures [20]. The high-entropy ceramic is a promising material for jet engine elements. To date, the problems of obtaining high-entropy borides remain poorly understood. Synthesis is carried out mainly using powder metallurgy technologies. A significant difference is observed only in the starting materials for synthesis. Most of the work begins with wet grinding and mixing of precursors in mills.

One of the first attempts to use HEB as an ultra-high temperature ceramic was made by Guild et al. [2]. The authors synthesized six high-entropy boride compositions with an AlB_2_-type structure: (Hf_0.2_Zr_0.2_Ta_0.2_Nb_0.2_Ti_0.2_)B_2_, (Hf_0.2_Zr_0.2_Ta_0.2_Mo_0.2_Ti_0.2_)B_2_, (Hf_0.2_Zr_0.2_Mo_0.2_Nb_0.2_Ti_0.2_)B_2_, (Hf_0.2_Mo_0.2_Ta_0.2_Nb_0.2_Ti_0.2_)B_2_, (Mo_0.2_Zr_0.2_Ta_0.2_Nb_0.2_Ti_0.2_)B_2_ and (Hf_0.2_Zr_0.2_Ta_0.2_Cr_0.2_Ti_0.2_)B_2_. The alloys were synthesized using a ball mill and spark plasma sintering (SPS). The research results showed that all these HEBs have higher oxidation resistance and higher hardness compared to the corresponding binary borides. The authors found that the oxidation process depends not only on the composition, but also on the microstructure and method of synthesis. High-entropy borides in most studies are represented by metals of the fourth subgroup (Ti, Zr, Hf) and fifth subgroup (V, Nb, Ta) of the periodic table.

Mechanical properties are also important aspects of experimental exploration, which are the essential references for engineering design and application of structural ceramics [21]. Some investigations have proved that high entropy is an effective way to improve fracture toughness. In these studies, in most cases, mechanical alloying and subsequent spark plasma sintering were used to obtain HEBs [22]. Zhu et al. studied the phase structure and properties of (Hf_0.25_Zr_0.25_Ta_0.25_Sc_0.25_)B_2_ HEB through the first-principle method combined with experiments. It was found that Sc has a significant impact on the internal bonding and properties of HEBs [23].

The main challenge for the synthesis of high-entropy boride ceramics is full densification. To achieve densification at a moderate temperature and avoid abnormal grain growth, it is necessary to enhance the sinterability of high-entropy boride powder. Tallarita et al. [24] synthesized (Hf_0.2_Mo_0.2_Zr_0.2_Nb_0.2_Ti_0.2_)B_2_ by reactive spark plasma sintering and studied the oxidative behavior by dynamic and isothermal thermogravimetric analysis in air. The material showed high resistance to oxidation at temperatures up to 1473 K for 6 h. The strong force between chemical bonds plays a crucial role in the crystal structure stability of high-entropy ultra-high temperature borides. The oxidation stability of HEB is equal to that of HfB_2_ and is better than that of other binary borides.

Materials with only equiatomic composition were investigated in the cited works. This narrows the operating temperature range of the material due to the formation of unstable and fusible oxides (Nb, Ta, Ti). The solution to this problem can be the creation of high-entropy ceramics with a high content of Hf and Zr.

This paper presents the results of studies on the preparation of single-phase high-entropy borides with a high degree of chemical homogeneity using mechanically alloyed (MA) powders of the TiZrHfNbTa system. The high-energy ball milling was utilized to reduce the particle size and form the solid solution. The effect of various synthesis parameters on the phase composition and distribution of elements is considered. The stability of the synthesized high-entropy borides in a high-temperature oxidizing flow has been studied.

## 2. Materials and Methods

Elemental powders of metals Ti, Nb, Hf, Zr and Ta (purity 99.5%) were used as initial components for the synthesis of the TiZrHfNbTa alloy. Two compounds were synthesized in experimental studies: TiZrHfTaNb and (TiTaNb)_0.3_Hf_0.35_Zr_0.35_. MA was carried out using a Fritsch Pulverisette 4 planetary mill in an argon atmosphere; MA duration 5–10 h; rotation speed of the planetary disk/pots—200–400 rpm; pots material—high-strength steel; glass volume—500 mL; grinding balls material—high-strength steel; ball diameter—12 mm; sample weight—50 g; and the ratio of powder weight to balls weight is 1:20.

The particle size distribution of the powders was analyzed using a laser diffraction unit (Fritsch Analysette 22 NanoTec plus, Idar-Oberstein, Germany). The Fraunhofer model was used to calculate the particle size distribution.

High-entropy alloy powder after mechanical alloying was mixed with polycrystalline boron powder for the synthesis of boride in a molar ratio of 1:2. Sintering was carried out using an SPS unit (HPD 25 FCT Systeme GmbH, Effelder-Rauenstein, Germany) in a graphite mold Ø20 mm, at temperatures of 1600, 1800, and 2000 °C, a pressure of 50 MPa, and the holding time at the maximum temperature was 5 min.

The phase composition of the powder was studied by a diffractometer (Bruker D8 Advance Cu Kα = 1.5418 Å) in the angle range of 20–100° with steps of 0.02° and a constant counting time of 1 s. Refinement of structural parameters and quantitative phase analysis were carried out by the Rietveld method (fundamental parameters) using the TOPAS 5 program. A layer-by-layer study of the phase composition of the samples after gas-dynamic tests was carried out by X-ray diffractometer (Rigaku SmartLab CuKα = 1.5418 Å) using CBO-µ confocal microfocus optics in grazing incidence mode (ω = 10°). The range of angles was 20–80° with a scan speed of 0.2° per minute. The particle morphology, microstructure, chemical composition and EDX mapping of the resulting powders were studied by a scanning electron microscope (Mira 3 Tescan, Brno, Czech Republic) with an EDX Oxford Instruments X-Max 80 energy-dispersive detector.

Microhardness was measured by a Buehler microhardness tester at 300 and 500 g. The measurements were carried out on ground and polished samples along a section parallel to the height of the cylindrical sample. The measurements were carried out in a straight line with a step of 330 μm from the upper edge of the sample to the lower one.

Gas-dynamic tests were carried out on an electric arc plasma torch UPIM-200 JSC “Composite”, plasma-forming gas–air. The heat flux was determined by a non-stationary method on a cold copper barrier (calorimeter). The heat flux density was 3.11 MW/m^2^, with a stepwise increase in power and an increase in the heat flux density by 0.4 MW/m^2^ at each stage. During testing, the outer temperature of the sample surface was recorded using a pyrometer, and the temperature distribution over the surface using a thermal imager.

## 3. Results

At the initial stages of mechanical alloying, the dissolution of alloying elements in all the studied systems is of a general nature. The particles of the original powder are flattened and welded together due to severe plastic deformation. The resulting composite particles have a layered structure of 5 h MA. Homogenization by chemical composition occurs with a further increase in the MA time to 7.5 h (see the Appendix A). This is due to a decrease in the free energy of the system (Figure 1). With a further increase in the MA time to 10 h, the content of technical iron grinding increases significantly. The particle size distribution was: d_10_ = 19.3 ± 0.6 µm, d_50_ = 47.5 ± 0.9 µm, d_90_ = 87.9 ± 1.3 µm (mean 52.4, span 1.44) for the sample after 5 h of MA; d_10_ = 8.2 ± 0.2 µm, d_50_ = 18.6 ± 0.4 µm, d_90_ = 33.8 ± 0.6 µm (mean 20.5 µm, span = 1.37) for sample after 7.5 h of MA; d_10_ = 17.1 ± 0.7 µm, d_50_ = 33.6 ± 0.8 µm, d_90_ = 59.3 ± 1.2 µm (mean 37.3 µm, span 1.25) for the sample after 10 h of MA. The asymmetry is probably due to the incompleteness of the powder grinding process.

Figure 2 shows X-ray patterns of the initial mixture of powders and powders after mechanical alloying for 5, 7.5, and 10 h. A decrease in the size of the coherent scattering regions (CSR) and an increase in microstresses in the material leads to a broadening of the peaks in the X-ray pattern. After 5 h of MA, complete dissolution of Ti in the bcc lattice of Nb and Ta is observed. This is explained by the close atomic radii of these elements (Ti = 1.45 Å, Nb = 1.43 Å, Ta = 1.43 Å). After 10 h of mechanical alloying, small Zr and Hf peaks are still observed, which is caused by their large atomic size.

The mass fraction of cubic (*Im*-3*m*) and hexagonal (*P*6_3_/*mmc*) phases was 83% and 17%, respectively. The crystal lattice parameter of the cubic structure was 3.387 Å. The deviation from the lattice parameter of the cubic structure calculated according to the Vegard law (*a* = 3.416 Å) for the equiatomic TiZrHfTaNb is due to the incomplete dissolution of the elements Hf and Zr.

Analysis of the microstructure, phase composition, distribution of elements and granulometric composition showed that after 7.5 h of mechanical alloying in the mode of 200–400 rpm (speed of rotation of the planetary disk and pots), the powder has a uniform distribution of elements and a low value of iron grinding. This mode was chosen for the synthesis of borides.

A mechanical mixture of high entropy alloy powder and polycrystalline boron was used to synthesize borides. The morphology of the mechanically alloyed TiZrHfTaNb powder is a flake–lamellar mixture. This form of particles was formed because of plastic deformation and adhesion of particles under the impact of balls. The synthesis of high-entropy borides was carried out using mechanically pre-alloyed powders by FCT HPD 25 spark plasma sintering unit.

Experimental data obtained during the synthesis of high-entropy boride (TiZrHfTaNb)B_2_ (Figure 3) indicate the occurrence of four main stages of synthesis: (1) the stage of degassing and evacuation; (2) the stage of preliminary sintering; (3) the stage of chemical interaction of metal–boron and the formation of high-entropy boride; and (4) compaction under pressure of 50 MPa with homogenization at a holding time of 5 min.

In the first stage, the mold is heated and evacuated. The volume that the powder occupies in the mold decreases due to the release of gases. Therefore, we see an increase in the shrinkage rate. In the second stage, the mold is heated without shrinkage. The third stage for samples sintered at different temperatures starts at 1300 °C and ends at 1600 °C. It can be seen from the results of the phase analysis (Figure 4) that during sintering at a temperature of 1600 °C, MeB_2_ is already formed. In connection with this, we assume that the chemical reaction Me + 2B → MeB_2_ proceeds at the third stage. In the fourth stage, the processes of plastic deformation and sintering of the powder with pressure take place. For this reason, an increase in the shrinkage rate of the sample is observed.

The study of the phase composition of the (TiZrHfTaNb)B_2_ boride samples (Figure 4) sintered at temperatures of 1600 °C, 1800 °C, and 2000 °C indicates the formation of a hexagonal structure of the MeB_2_ type (*P*6_3_/*mmc*). The use of graphite tooling leads to partial carburization of the surface layers with the formation of hafnium-tantalum carbide with the formula (HfTa)C. The presence of a halo and its intensity in X-ray patterns depend on the synthesis conditions. During the reaction sintering of samples without applied external pressure, there is no significant compaction of the mixture. This causes the presence of a halo in the diffraction pattern (13) from the X-ray amorphous fine-grained ceramic component. The diffraction patterns show low-intensity peaks corresponding to ZrO_2_ and HfO_2_ oxides.

Figure 5 shows the microstructure and distribution of elements obtained using SEM-EDS mapping. The results confirmed the presence of zirconium and hafnium oxides. The distribution and shape of inclusions of hafnium oxide (white inclusions) and zirconium oxides (light gray inclusions) essentially represent point submicron inclusions located in a line or clusters, presumably along the boundaries of the formed boride grains. The formation of high-entropy borides (stage 3) during spark plasma sintering is accompanied by large heat releases. The described kinetics of chemical reactions for the formation of high-entropy borides refers to self-propagating high-temperature synthesis. The calculated values of the temperature distribution during sintering are shown in Figure 6. Figure 6 shows the distribution of temperatures (Figure 6b) and stresses (Figure 6c–e) during spark plasma sintering in the mold. The highest temperature is reached in the center of the upper part of the sample. In this place, the process of self-propagating high-temperature synthesis is induced and propagates in the radial and axial directions. During the combustion of the powder mixture, the particles melt, and redox reactions take place in parallel. As a result of these processes, inclusions of hafnium and zirconium oxides are formed, due to the lowest free energy (16) of the formation of oxides of these metals.

An increase in the sintering temperature of high-entropy boride (TiZrHfTaNb)B_2_ to 1800 °C and 2000 °C leads to homogenization of the chemical and phase composition due to coagulation of hafnium and zirconium oxide inclusions and a decrease in their total content (Figure 7 and Figure 8). An increase in temperature accelerates the diffusion of heavy elements, which contributes to the homogenization of the sample. The content of nonequilibrium inclusions in these samples is lower compared to the sample sintered at a temperature of 1600 °C. The intensity of the oxide peaks decreases with increasing temperature. The halo on the diffractogram after sintering at 1800 °C is preserved. Its intensity does not change, which is associated with the content of the X-ray amorphous fine-grained ceramic component [13]. The decrease in oxide phases in the sample obtained by sintering at 2000 °C may be due to the formation and evaporation of boron oxide (B_2_O_3_) at temperatures above 1860 °C. And this may also be due to the interaction of metal oxides with tooling carbon at temperatures around 2000 °C. Increasing the temperature or holding time during sintering will likely lead to the complete dissolution of the oxide phases. The absence of a halo in the diffraction pattern (Figure 4) indicates the completion of the sintering process and the dissolution of the X-ray amorphous fine-grained component.

The results of microhardness measurements (Figure 9) confirmed the studies of the microstructure and phase composition. The average hardness of the samples is 1959, 1707, and 1763 HV, respectively, which is comparable to previous studies [2]. A sample sintered at 1600 °C has a higher hardness. High hardness is caused by stresses of the second kind due to the presence of non-equilibrium oxide, boride, and transition phases. Increasing the sintering temperature led to the formation of a single-phase high-entropy boride with a more equilibrium structure. This is the reason for the decrease in hardness when the sintering temperature increases to 1800 °C. With a further increase in the sintering temperature to 2000 °C, the hardness increases slightly.

The synthesis of HEA-based boride (TiTaNb)_0.3_Hf_0.35_Zr_0.35_ was carried out using similar sintering conditions at a temperature of 2000 °C. As a result of the complex experimental studies performed, single-phase boride (TiTaNb)_0.3_Hf_0.35_Zr_0.35_B_2_ was synthesized. The microstructure and phase composition of the sample are similar to the material (TiTaNbHfZr)B_2_.

## 4. Discussion

Figure 10 shows the dependence of the surface temperature of samples of high-entropy borides in a high-temperature oxidizing gas flow on the test time. The heat flux density for the selected mode was 3.11 MW/m^2^ for 135 s (including the time to enter the mode); 3.47 MW/m^2^ for 60 s; 3.83 MW/m^2^ for 100 s; and 4.2 MW/m^2^ for 100 s or until failure/melting. The initial power was chosen approximately because there were no data on the possible characteristics of the material. The duration of testing a sample of equiatomic composition was 215 s.

The liquid phase appeared on the surface of the sample at a temperature of about 1815 °C (Figure 11). The maximum test temperature of the sample without intensive entrainment and melting was 1904 °C. The duration of testing the (TiTaNb)_0.3_Hf_0.35_Zr_0.35_B_2_ sample was 330 s. The liquid phase appeared on the surface of the sample at the second power level at a temperature of about 1910 °C (Figure 12). Further, when the power was increased, the intensity of the formation of the liquid phase increased. The maximum test temperature of the sample without intensive entrainment and melting was 2070 °C.

Layer-by-layer phase analysis revealed the formation of oxide layers with variable phase composition (Figure 12). The surface layer is represented by mixed refractory MeO_2_ dioxide (prototype ZrO_2_, T_melt_ = 2715 °C). The second layer consists of a mixture of MeB_2_ diboride and MeO_2_ dioxide. The third layer consists of a mixed low-melting oxide Me_4_O_5_ (prototype Nb_4_O_5_, T_m_ = 1900 °C), MeB_2_ diboride, and MeO_2_ dioxide. The surface of the sample is represented by a mixed fused dioxide with the formula MeO_2_ in which there are pores from the evaporation of low-melting metal oxides and boron oxide. The results of the analysis confirmed the formation of layers enriched in Zr and Hf, Nb, and Ta (Figure 12 and Figure 13).

## 5. Conclusions

The paper presents the results of obtaining single-phase high-entropy ceramic materials with a high degree of chemical homogeneity using the equiatomic high-entropy boride (TiZrHfTaNb)B_2_ as an example. Single-phase and chemically homogeneous high-entropy borides were obtained by preliminary mixing of metals at the atomic level.

The results of process research of mechanical alloying of TiZrHfTaNb showed that a homogeneous chemical composition with the lowest iron content (0.12%) was achieved with the following mechanical alloying modes: a rotation of the planetary disk/pots was 200–400 rpm and a process time was of 7.5 h. During sintering at temperatures around 1600 °C, a high-entropy boride with the chemical formula MeB_2_ is formed. Inclusions in the form of zirconium-hafnium oxide and a transition layer between oxide inclusions and the main boride phase are formed on the surface.

Increasing the process temperature to 2000 °C leads to the formation of single-phase and chemically homogeneous high-entropy boride (TiZrHfTaNb)B_2_. The decrease in the oxide phases in the sample obtained by sintering at 2000 °C can be associated with the formation and evaporation of boron oxide (B_2_O_3_) at temperatures above 1860 °C, as well as the activation of the interaction of metal oxides with tooling carbon at temperatures of about 2000 °C, similarly to the oxidation–reduction reactions occurring during the synthesis of high-entropy borides.

Tests of the synthesized materials in a high-temperature oxidizing gas flow showed that with an increase in the content of Zr and Hf in the composition, the temperature of the formation of the liquid phase on the surface of the sample increases; therefore, the maximum temperature without intensive entrainment and the potential operating time of the material increase.

The proposed method for the synthesis of high-entropy ceramics has the following advantages: it makes it possible to obtain materials that are single-phase and homogeneous in chemical composition, the minimum content of impurity phases, a controlled synthesis process, repeatability of results and the possibility of scaling.

## Figures and Tables

**Figure 1 materials-16-06744-f001:**
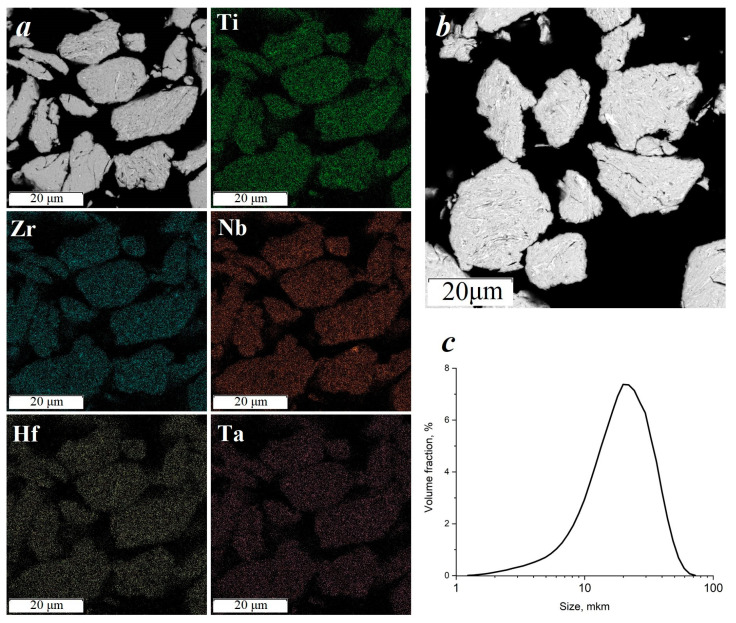
TiZrHfTaNb powder after 7.5 h of mechanical alloying: distribution of elements (**a**), powder microstructure (**b**), particle size distribution (**c**).

**Figure 2 materials-16-06744-f002:**
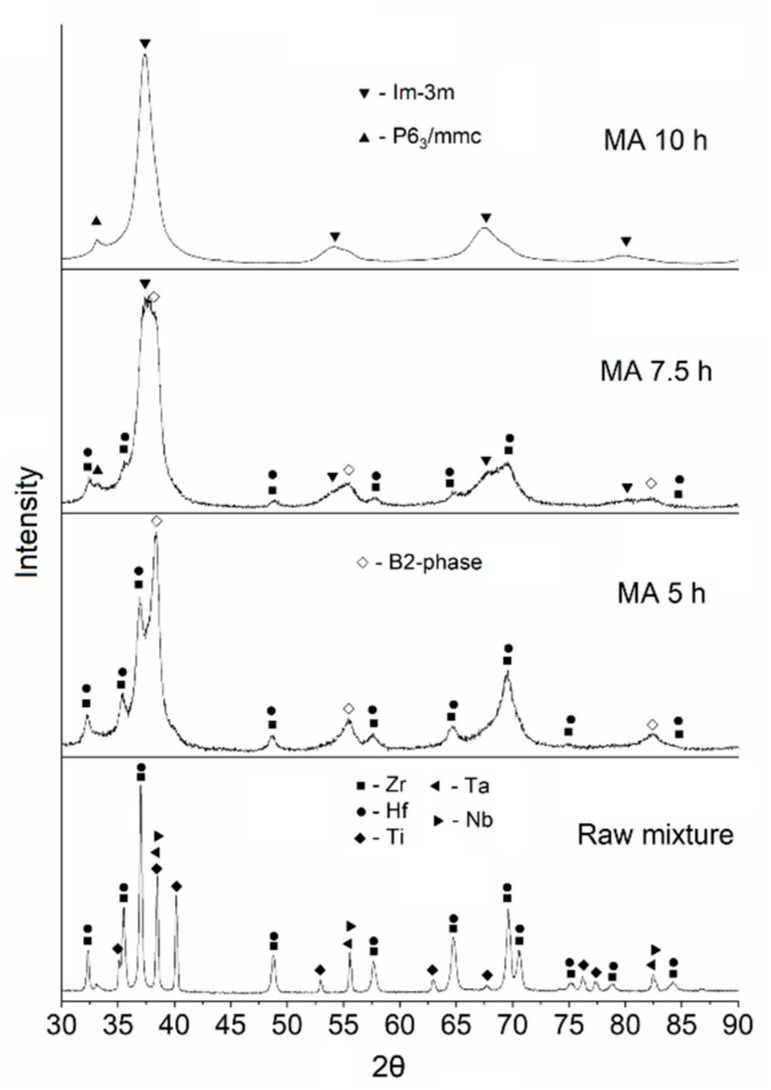
Phase composition of TiZrHfTaNb powder after mechanical alloying.

**Figure 3 materials-16-06744-f003:**
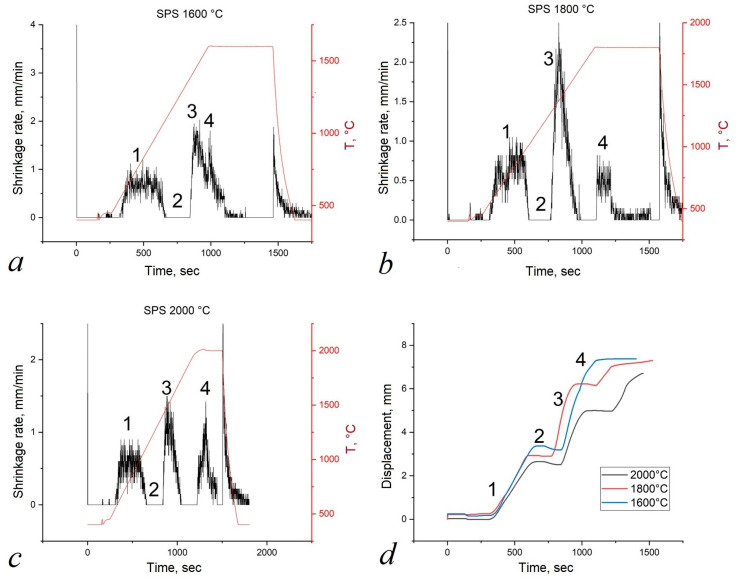
Experimental curves obtained during the synthesis of (TiZrHfTaNb)B_2_: (**a**) sintering at 1600 °C; (**b**) sintering at 1800 °C; (**c**) sintering at 2000 °C; (**d**) time travel. Stages of synthesis: (1) degassing and evacuation; (2) preliminary sintering; (3) chemical interaction; (4) compaction.

**Figure 4 materials-16-06744-f004:**
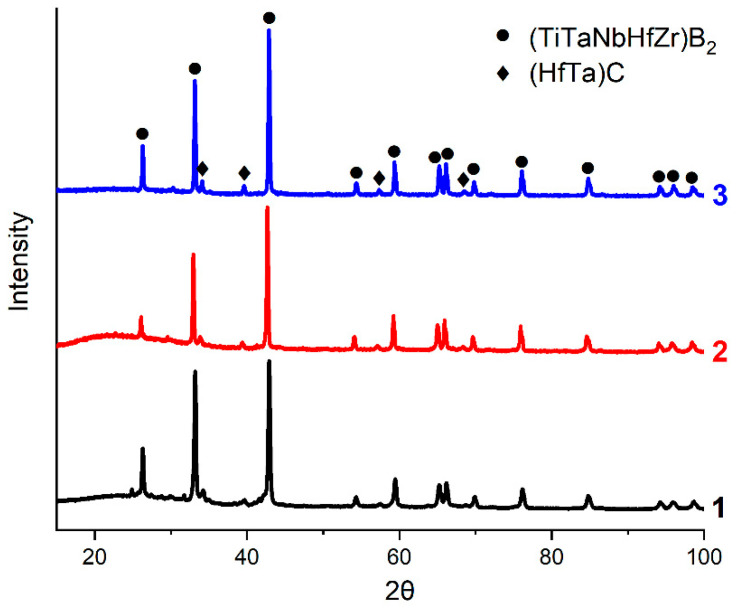
X-ray pattern of high-entropy borides (TiZrHfTaNb)B_2_ sintered at temperatures: 1—1600 °C, 2—1800 °C, 3—2000 °C.

**Figure 5 materials-16-06744-f005:**
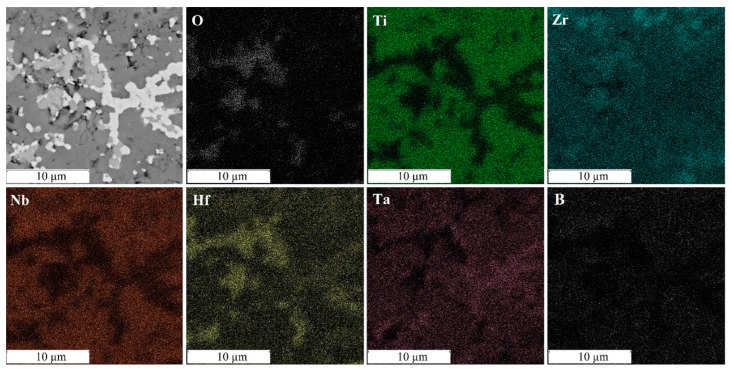
Microstructure and distribution of elements of the sample (TiZrHfTaNb)B_2_, sintered at a temperature of 1600 °C.

**Figure 6 materials-16-06744-f006:**
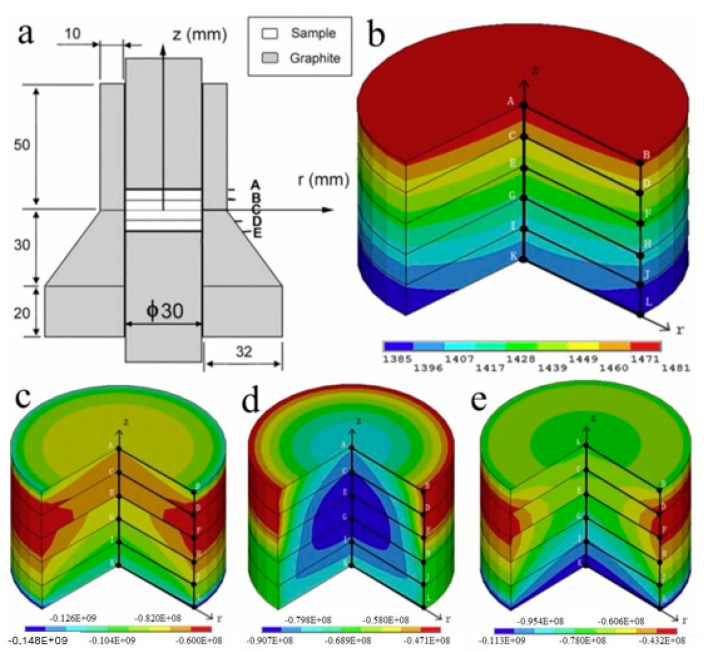
Temperature distribution and stress over the cross section (A–L) of the Ti-TiB sample made by the SPS method (180 °C/min, t = 360 s): (**a**) a schematic representation of the stamp; (**b**) temperature distribution; (**c**) radial stress distribution; (**d**) axial stress distribution; and (**e**) circumferential stress distribution.

**Figure 7 materials-16-06744-f007:**
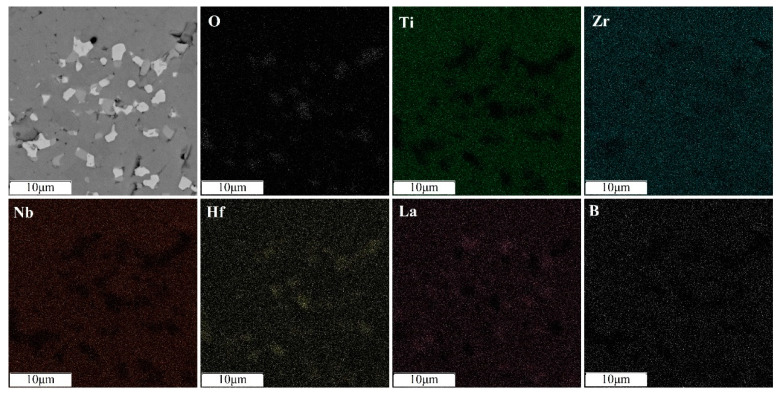
Distribution of elements of the (TiZrHfTaNb)B_2_ sample sintered at temperature of 1800 °C.

**Figure 8 materials-16-06744-f008:**
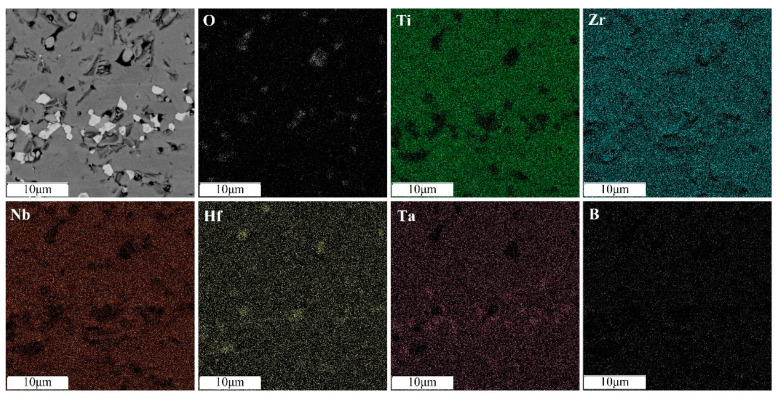
Distribution of elements of the (TiZrHfTaNb)B_2_ sample sintered at temperature of 2000 °C.

**Figure 9 materials-16-06744-f009:**
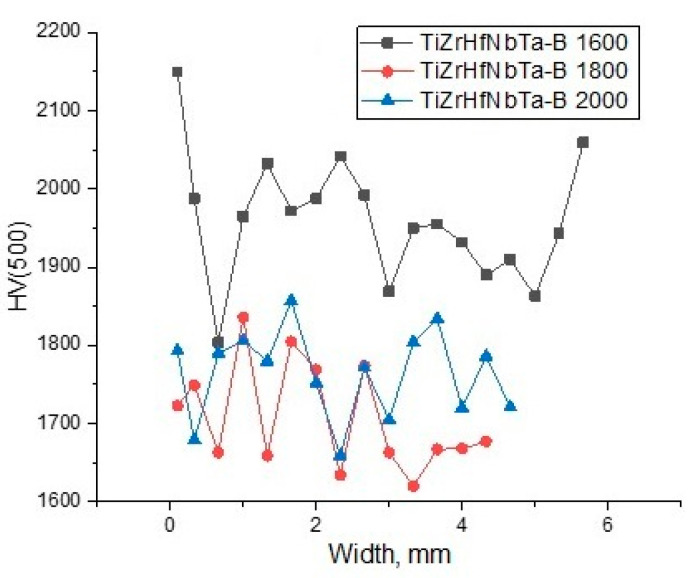
Microhardness in cross-section, sintered specimens (TiZrHfTaNb)B_2_.

**Figure 10 materials-16-06744-f010:**
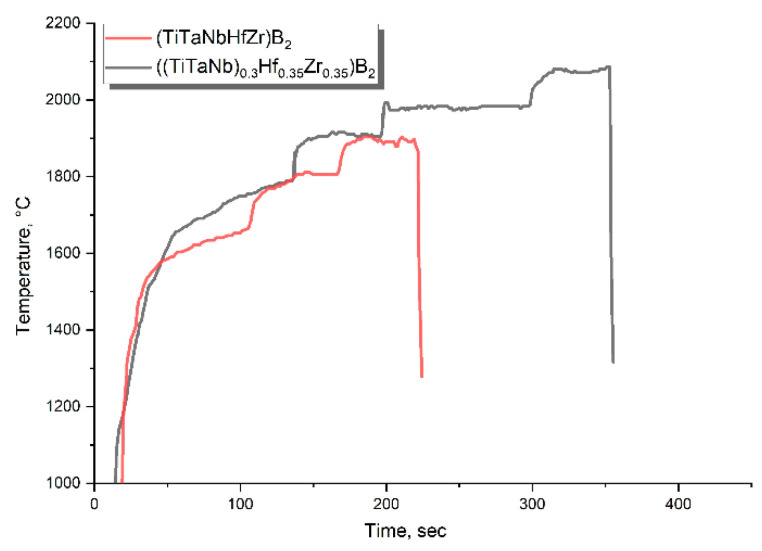
Dependence of the surface temperature of samples of high-entropy borides on the test time.

**Figure 11 materials-16-06744-f011:**
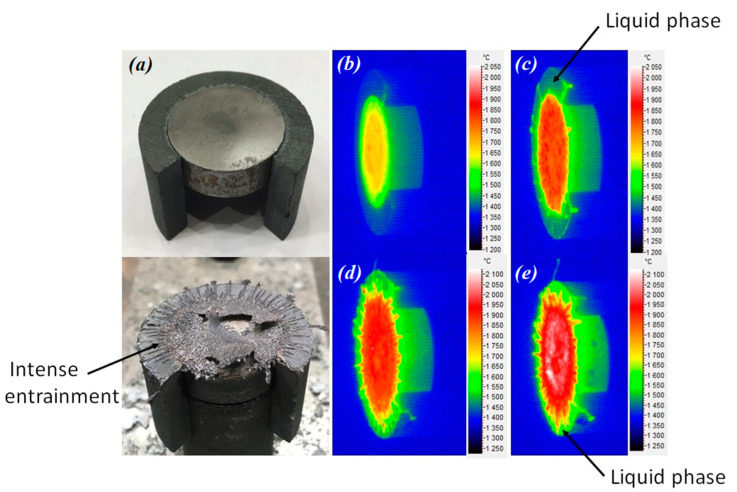
Appearance of the sample (TiTaNb)_0.3_Hf_0.35_Zr_0.35_B_2_ before/after gas-dynamic tests (**a**), sample test thermogram after: 120 s (**b**); 160 s (**c**); 250 s (**d**); 330 s (**e**).

**Figure 12 materials-16-06744-f012:**
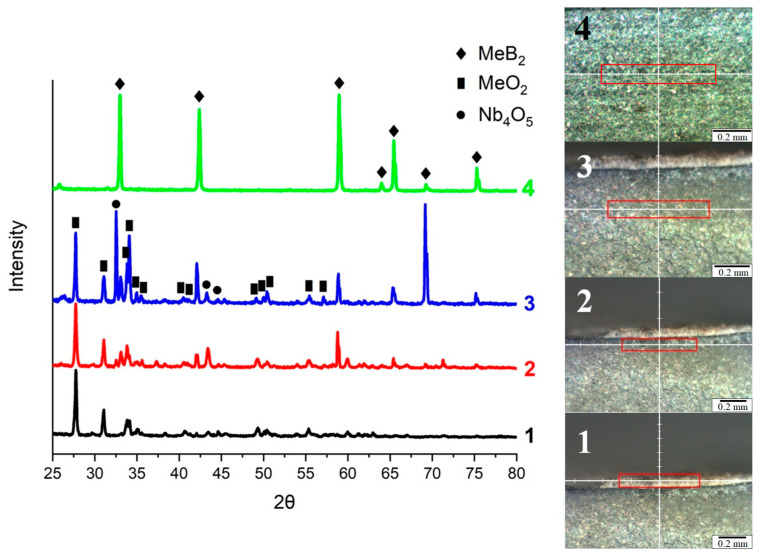
Layer-by-layer (1–4) study of the phase composition in the cross-section of the sample (TiTaNb)_0.3_Hf_0.35_Zr_0.35_B_2_.

**Figure 13 materials-16-06744-f013:**
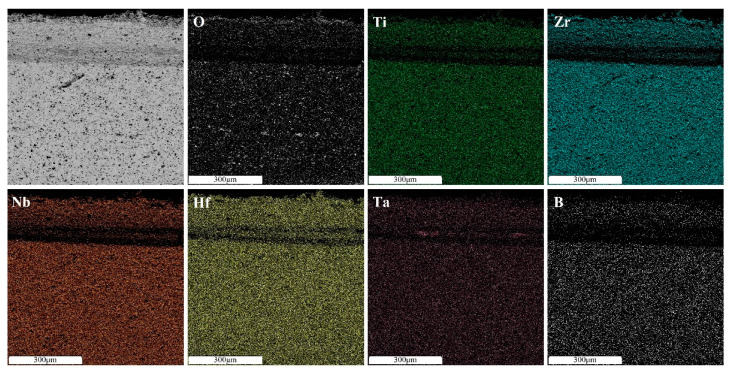
Distribution of elements in the cross-section of the (TiTaNb)_0.3_Hf_0.35_Zr_0.35_B_2_ sample by reaction products, transition zone, and base material.

## Data Availability

The data presented in this study are available upon request from the corresponding author. Data are not publicly available due to confidentiality.

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
