# Peer review of "Structure and Properties of High-Entropy Boride Ceramics Synthesized by Mechanical Alloying and Spark Plasma Sintering"

_materials, 2023, doi:10.3390/ma16206744_

Round 1

Reviewer 1 Report

Razumov et al. demonstrate the synthesis of high-entropy boride MeB2 (Me = Ti, Ta, Nb, Hf, Zr) via mechanical alloying and spark plasma sintering. A single-phase and homogeneous high-entropy ceramics is achieved at a high process temperature. Moreover, the mechanical and chemical properties of the synthesized high-entropy ceramics are systematically studied. The work is significant, and the results support the conclusion. Therefore, I recommend the publication of this paper in “Materials”.

1. In the abstract, the authors state that a hexagonal structure is formed for the high-entropy borides. However, there is no discussion about this in the paper.

2. Compared with the simple binary borides, does the high-entropy boride show some different or enhanced properties?

3. In the paper, the authors present the high-entropy carbides instead of high-entropy borides. Please check it.

Author Response

Authors are thankful to the reviewer for the valuable comments. Following are the reply to the comments. Necessary changes in the revised version of the manuscript have been incorporated.

1. In the abstract, the authors state that a hexagonal structure is formed for the high-entropy borides. However, there is no discussion about this in the paper.

Author’s response: Author apologizes for the lack of clarity. Changes have been made to the text (page 6 line 183)

2. Compared with the simple binary borides, does the high-entropy boride show some different or enhanced properties?

Author’s response: Author thanks the reviewer’s concern. According to previous studies, high-entropy borides have higher mechanical properties compared to simple binary borides and higher oxidation resistance. Information on this has been added to the text (page 2 line 57-73).

3. In the paper, the authors present the high-entropy carbides instead of high-entropy borides. Please check it.

Author’s response: Author apologizes for this mistake. The necessary changes have been made.

All changes made to the text are highlighted in yellow.

Reviewer 2 Report

The paper presents the results of obtaining single-phase high-entropy ceramic materials with a high degree of chemical homogeneity using the equiatomic high-entropy carbide (TiZrHfTaNb)B2. Totally speaking, this work is interesting, but some problems need to be addressed before acceptance.

1. The mass fraction of the hexagonal phase was 17%, and the crystal lattice parameter of the cubic structure was 3.387 Å . How did the authors get the data? The authors should give instruction.

2. Which type of crystal structure did the high-entropy TiZrHfTaNb powder and the high-entropy borides belong to? The authors should give the detailed statement.

3. At three different sintering temperatures, the average hardness of the samples is 1959, 1707 and 1763 HV, respectively. Why did the hardness decrease as the temperature increased? In addition, compared to other alloys, what is the hardness level of this alloy? The author must give a detailed statement.

4. The microstructure and phase composition of the sample is similar to the material (TiTaNbHfZr)B2. It is better to give supplementary materials to support this conclusion.

The paper presents the results of obtaining single-phase high-entropy ceramic materials with a high degree of chemical homogeneity using the equiatomic high-entropy carbide (TiZrHfTaNb)B2. Totally speaking, this work is interesting, but some problems need to be addressed before acceptance.

1. The mass fraction of the hexagonal phase was 17%, and the crystal lattice parameter of the cubic structure was 3.387 Å . How did the authors get the data? The authors should give instruction.

2. Which type of crystal structure did the high-entropy TiZrHfTaNb powder and the high-entropy borides belong to? The authors should give the detailed statement.

3. At three different sintering temperatures, the average hardness of the samples is 1959, 1707 and 1763 HV, respectively. Why did the hardness decrease as the temperature increased? In addition, compared to other alloys, what is the hardness level of this alloy? The author must give a detailed statement.

4. The microstructure and phase composition of the sample is similar to the material (TiTaNbHfZr)B2. It is better to give supplementary materials to support this conclusion.

Author Response

Authors are thankful to the reviewer for the valuable comments. Following are the reply to the comments. Necessary changes in the revised version of the manuscript have been incorporated.

1. The mass fraction of the hexagonal phase was 17%, and the crystal lattice parameter of the cubic structure was 3.387 Å . How did the authors get the data? The authors should give instruction.

Author’s response: Author apologizes for the lack of clarity. Added information about methods for processing diffraction patterns. (page 3 line 103)

"Refinement of structural parameters and quantitative phase analysis were carried out by the Rietveld method (Fundamental parameters) using the TOPAS 5 program."

2. Which type of crystal structure did the high-entropy TiZrHfTaNb powder and the high-entropy borides belong to? The authors should give the detailed statement.

Author’s response: The powder after mechanical alloying consists of two phases: hexagonal and cubic. After sintering, the boride has a hexagonal structure. Corrections have been made to the text.

(page 4 line 148)
"The mass fraction of cubic (Im-3m) and hexagonal (P63/mmc) phases was 83% and 17%, respectively."

(page 6 line 183)
The study of the phase composition of the (TiZrHfTaNb)B2 boride samples (Figure 4) sintered at temperatures of 1600 °C, 1800 °C and 2000 °C indicates the formation of a hexagonal structure of the MeB2 type (P63/mmc).

3. At three different sintering temperatures, the average hardness of the samples is 1959, 1707 and 1763 HV, respectively. Why did the hardness decrease as the temperature increased? In addition, compared to other alloys, what is the hardness level of this alloy? The author must give a detailed statement.

Author’s response: The text has been edited and the following text has been added:

(page 9 line 240)
"The average hardness of the samples is 1959, 1707 and 1763 HV, respectively, which is comparable to previous studies [2]. A sample sintered at 1600 °C has a higher hardness. High hardness is caused by stresses of the second kind due to the presence of non-equilibrium oxide, boride and transition phases. Increasing the sintering temperature led to the formation of a single-phase high-entropy boride with a more equilibrium structure. This is the reason for the decrease in hardness when the sintering temperature increases to 1800 °C. With a further increase in the sintering temperature to 2000 °C, the hardness increases slightly."

4. The microstructure and phase composition of the sample is similar to the material (TiTaNbHfZr)B2. It is better to give supplementary materials to support this conclusion.

Author’s response: Authors thank the reviewer’s suggestion. Mapping results of (TiTaNb)0.3Hf0.35Zr0.35B2 after 7.5 hours of mechanical alloying in the attachment

Changes made are highlighted in yellow.

Reviewer 3 Report

In this manuscript it is presented results about single-phase high-entropy ceramic materials with a high degree of chemical homogeneity using the equiatomic high-entropy carbide (TiZrHfTaNb)B2 as an example. The proposed method for the synthesis of high-entropy ceramics has the following advantages: it makes it possible to obtain materials that are single-phase and homogeneous in chemical composition, the minimum content of impurity phases, a controlled synthesis process, repeatability of results, and the possibility of scaling.

The results are interesting, and for this reason, the manuscript could be published if the authors address the following issues:

1) Regarding Figures 1, 5, 7, 8, and 13, the authors should provide detailed information about:

(a) The methodology used to obtain these figures;

(b) Whether any image processing techniques were applied.

2) The authors have presented a limited discussion regarding the particle size distribution in Figure 1(c). It is important to provide the statistical characteristics of this distribution, including the mean value, standard deviation, skewness, and so on. Furthermore, an explanation for the right-skewed distribution should be included.

3) The authors should engage in a more quantitative discussion of the homogenization process mentioned in this section of the text: 

"An increase in the sintering temperature of high-entropy boride (TiZrHfTaNb)B2 to 1800°C and 2000°C leads to the homogenization of the chemical and phase composition due to the coagulation of hafnium and zirconium oxide inclusions and a decrease in their total content (see Figure 7 and 8)."

4) In Figure 9, the authors should clarify why there is no significant statistical difference in microhardness between 1800°C and 2000°C."

Author Response

Authors are thankful to the reviewer for the valuable comments. Following are the reply to the comments. Necessary changes in the revised version of the manuscript have been incorporated.

1. Regarding Figures 1, 5, 7, 8, and 13, the authors should provide detailed information about:
(a) The methodology used to obtain these figures;
(b) Whether any image processing techniques were applied.

Author’s response: Author apologizes for the lack of clarity. The Figures were obtained by EDX mapping using a scanning electron microscope. No image processing techniques were used. In the "Materials and Methods" section the following text has been added:

"The particle morphology, microstructure, chemical composition and EDX mapping of the resulting powders were studied by a scanning electron microscope (Mira 3 Tescan) with an EDX Oxford Instruments X-Max 80 energy-dispersive detector."

2. The authors have presented a limited discussion regarding the particle size distribution in Figure 1(c). It is important to provide the statistical characteristics of this distribution, including the mean value, standard deviation, skewness, and so on. Furthermore, an explanation for the right-skewed distribution should be included.

Author’s response: Authors thank the reviewer’s suggestion. Text supplemented with data (Page 3).

3. The authors should engage in a more quantitative discussion of the homogenization process mentioned in this section of the text: 

"An increase in the sintering temperature of high-entropy boride (TiZrHfTaNb)B2 to 1800°C and 2000°C leads to the homogenization of the chemical and phase composition due to the coagulation of hafnium and zirconium oxide inclusions and a decrease in their total content (see Figure 7 and 8)."

Author’s response: Authors thank the reviewer’s suggestion. Information has been added to the text (page 8, line 219-234)

4. In Figure 9, the authors should clarify why there is no significant statistical difference in microhardness between 1800°C and 2000°C."

Author’s response: Author thanks the reviewer’s concern. Increasing the sintering temperature usually leads to an increase in hardness [10.1016/j.ijlmm.2022.01.002], [10.1016/j.matchar.2004.06.007]. However, in our alloy, on the one hand, an increase in temperature first reduces the hardness due to the dissolution of nonequilibrium phases, and a further increase increases the hardness due to the higher density of the material. Therefore, the difference is not a significant statistical difference in hardness.

Changes made are highlighted in yellow.